# Predicting Monthly Runoff of the Upper Yangtze River Based on Multiple Machine Learning Models

Xiao Li [1], Liping Zhang [1,2,*], Sidong Zeng [1,3], Zhenyu Tang [1], Lina Liu [1], Qin Zhang [1], Zhengyang Tang [4,*] and Xiaojun Hua [4]

1   State Key Laboratory of Water Resources and Hydropower Engineering Science, Wuhan University, Wuhan 430072, China
2   Institute for Water-Carbon Cycles and Carbon Neutrality, Wuhan University, Wuhan 430072, China
3   Chongqing Institute of Green and Intelligent Technology, Chinese Academy of Sciences, Chongqing 400714, China
4   Hubei Key Laboratory of Intelligent Yangtze and Hydroelectric Science, China Yangtze Power Co., Ltd., Yichang 443000, China
*   Correspondence: zhanglp@whu.edu.cn (L.Z.); tang_zhengyang@ctg.com.cn (Z.T.)

**Abstract:** Accurate monthly runoff prediction is significant to extreme flood control and water resources management. However, traditional statistical models without multi-variable input may fail to capture runoff changes effectively due to the dual effect of climate change and human activities. Here, we used five multi-input machine learning (ML) models to predict monthly runoff, where multiple global circulation indexes and surface meteorological indexes were selected as explanatory variables by the stepwise regression or copula entropy methods. Moreover, four univariate models were adopted as benchmarks. The multi-input ML models were tested at two typical hydrological stations (i.e., Gaochang and Cuntan) in the Upper Yangtze River. The results indicate that the LSTM_Copula (long short-term memory model combined with copula entropy method) model outperformed other models in both hydrological stations, while the GRU_Step (gate recurrent unit model combined with stepwise regression method) model and the RF_Copula (random forest model combined with copula entropy method) model also showed satisfactory performances. In addition, the ML models with multi-variable input provided better predictability compared with four univariate statistical models, and the *MAPE* (mean absolute percentage error), *RMSE* (root mean square error), *NSE* (Nash–Sutcliffe efficiency coefficient), and *R* (Pearson's correlation coefficient) values were improved by 5.10, 4.16, 5.34, and 0.43% for the Gaochang Station, and 10.84, 17.28, 13.68, and 3.55% for the Cuntan Station, suggesting the proposed ML approaches are practically applicable to monthly runoff forecasting in large rivers.

**Keywords:** monthly runoff prediction; machine learning; copula entropy; stepwise regression; Upper Yangtze River





## 1. Introduction

Flood is a complex interplay of hydrology, climate, and human management, and it is destructive to infrastructure, agriculture, and socioeconomic systems [1,2]. Yet water resource management, such as water conservation, flood control, and reservoir operation, relies heavily on accurate streamflow prediction [3–5]. In addition, streamflow is affected by multiple variables, such as precipitation, surface temperature, solar radiation, and atmospheric circulation, presenting the compound characteristics of strong nonlinearity, high uncertainty, and spatiotemporal variability [6–9]. Consequently, the high accuracy of monthly flood and streamflow prediction involving multiple impact factors has been emphasized urgently.

Different runoff prediction models have been proposed. Generally, these models can be divided into process-driven and data-driven models [5,10]. Process-driven models

based on the physical conception have some modeling conventions, such as data limitations and uncertainty for the initial conditions, process parameterizations, and computational constraints [11,12]. Among the data-driven models, traditional univariate statistical models only require runoff input [13], such as the autoregressive moving average (ARMA) model, mean generating function (MGF), nearest neighbor bootstrapping regressive (NNBR) model, and grey model (GM) [14–17]. Statistical models have been used extensively to capture the stationary and linear links in time series but may not be appropriate for predicting nonstationary and nonlinear runoff. Machine learning (ML) models [3], which are outstanding at handling nonlinear data, have been widely applied in hydrological prediction, including artificial neural networks (ANNs), support vector machine (SVM), gradient boosted decision tree (GBDT), etc. Derived from the biological neurons, ANNs build mapping with a vast amount of parameters to match the mapping between observation and prediction. Many new ANNs have arisen to provide more satisfactory solutions to time series forecasting problems, including the long short-term memory model (LSTM) and gate recurrent unit model (GRU). Due to the weakness of the conventional ANN for long-term dependencies, LSTM is constructed to improve the defects [18]; GRU has a much simpler structure and thus needs less calculation than LSTM. GBDT and random forests (RF) have been proven to be distinguished predictive models in many regression tasks with an ensemble of decision trees [19,20]. The SVM, founded on the theory of statistical learning as well as the principle of structural risk minimization [21], can address problems with limited samples, nonlinearity, and impartibility in low-dimensional space.

Unlike traditional regression models, ML models have considerably upgraded the mid-to-long-term forecasting performance of highly nonlinear streamflow time series. Many previous studies [5,11,13,22] have shown that the traditional statistical models' performances stagnated at 15–20% in terms of mean absolute percentage error, demonstrating the urgency of adopting new ML models. However, since there is no absolute optimal model for forecasting monthly runoff, it is necessary to use multiple ML models and compare their performances.

The identification of input variables is a critical part of the data-driven models. It is difficult to determine a suitable set of model inputs because of the complexity of the causes and the time lag in the runoff response to large-scale atmospheric circulation and surface meteorological variables [23]. Many teleconnection climate indexes [3] have been considered as alternative candidate predictive variables, including atmospheric circulation index, SST indexes, and other indexes, such as the total sunspot number index, Pacific decadal oscillation index, North Atlantic triple Index, etc. Besides, the antecedent runoff and other surface meteorological factors, such as precipitation, air pressure, temperature, wind speed, etc., are firmly connected on the grounds of physical links [24]. The methods used in variable selection are principally the correlation coefficient method, stepwise regression analysis, principal component analysis (PCA) [4], mutual information (MI), and partial mutual information (PMI) [25,26]. Among the various variable selection methods, the correlation coefficient method and the stepwise regression method are widely applied due to their simplicity and clarity [23]. Furthermore, copula entropy, a novel entropy concept defined by Ma and Sun [27,28] in 2008, is able to calculate the full-order correlations among variables and handle the redundant inputs directly. Copula has been applied in multivariate modeling with joint distributions, where two divisions, entropy copula and copula entropy, have been broadly employed in the hydrological study [28,29]. The entropy copula is mainly used in constructing a dependence structure with marginal probability distribution constraints. In addition, the copula entropy, which is outstanding in dependence analysis, has been used to study the variability of climate and hydrological variables, mostly precipitation, temperature, and streamflow [30]. However, the copula entropy method is rarely applied in nonlinear dependence measurement among large-scale circulations and streamflow in monthly runoff prediction, highlighting the importance of incorporating this method into data-driven model input variable identification [31].

The climate in the Upper Yangtze River Basin (UYRB) in China is complex due to complex topography and the interplay among diverse circulation systems (for instance, the East Asia monsoon, Indian monsoon, Australian monsoon, mid-latitude westerlies, and plateau monsoon) as well as the water conservancy projects construction [32]. The Three Gorges Reservoir (TGR), situated in the UYRB, is the world's largest hydropower station in terms of installed capacity [33–35], and the power production and the overall profits of which are heavily reliant on the upstream flows. Located in the lower reaches of the confluence of the Dadu River and Min River [36,37], the Gaochang Station is the outlet of the Min River basin, controlling 13.46% of the drainage area and 19.86% of the annual flow in the UYRB [38]. The Cuntan Station is located in Chongqing City, where the Jialing River meets the Yangtze River [39], and it controls about 60% of the water in the UYRB. Therefore, the monthly runoff prediction study on the Gaochang Station and Cuntan Station is not only related to the operation of TGR but also vital to human life and property in the middle and lower reaches.

In this study, we aimed to improve the accuracy of monthly streamflow prediction in the UYRB for the power generation in the Three Gorges Reservoir and the long-term flood management in the middle and lower reaches. Furthermore, the role of multi-variable inputs and the choice and performance of different predictive models are of particular interest to us. Therefore, five multi-input ML models were employed to make the best use of available data, combined with the stepwise regression method or copula entropy method to select input variables with different time lags. The ML models were carried out in two typical hydrological stations of the UYRB, and the detailed case study is presented in the following sections.

## 2. Methods

### 2.1. Study Area and Data

The mainstream of the UYRB is 4529 km long, and it charges a watershed of 1,000,000 km$^2$, which covers the region from 24.30° N–35.45° N to 90.33° E–112.04° E and represents 58.9% of the entire region of the Yangtze River [39,40]. Most of the regions in the UYRB are warm and moist, influenced by subtropical monsoon [24]. The mean annual streamflow varies between 700 and 2400 m$^3$/s, and the Three Gorges Station in the mainstream reached an annual runoff of 16,427 m$^3$/s in 1965, and the Zipingpu Station in the Min River was less than 265 m$^3$/s in 2006 [41]. Here, two critical hydrological stations—Gaochang and Cuntan in the UYRB were examined (Figure 1).

Monthly streamflow data of the Gaochang and Cuntan stations from January 1961 to December 2018 were collected in our study. One hundred and thirty monthly global circulation indexes within the same timeframe as the streamflow observations were downloaded from the National Climate Center of the China Meteorological Administration (http://cmdp.ncc-cma.net (accessed on 20 June 2022)), including 88 atmospheric circulations, 26 SST indexes, and 16 other indexes. Moreover, two meteorological stations near the selected hydrological stations were considered with complete monthly observations spanning the period of 1961–2018, including air pressure, average temperature, maximum temperature, minimum temperature, relative humidity, wind speed, and daylight hours, which were supplied by the China Meteorological Data Network (https://data.cma.cn/ (accessed on 20 June 2022)).

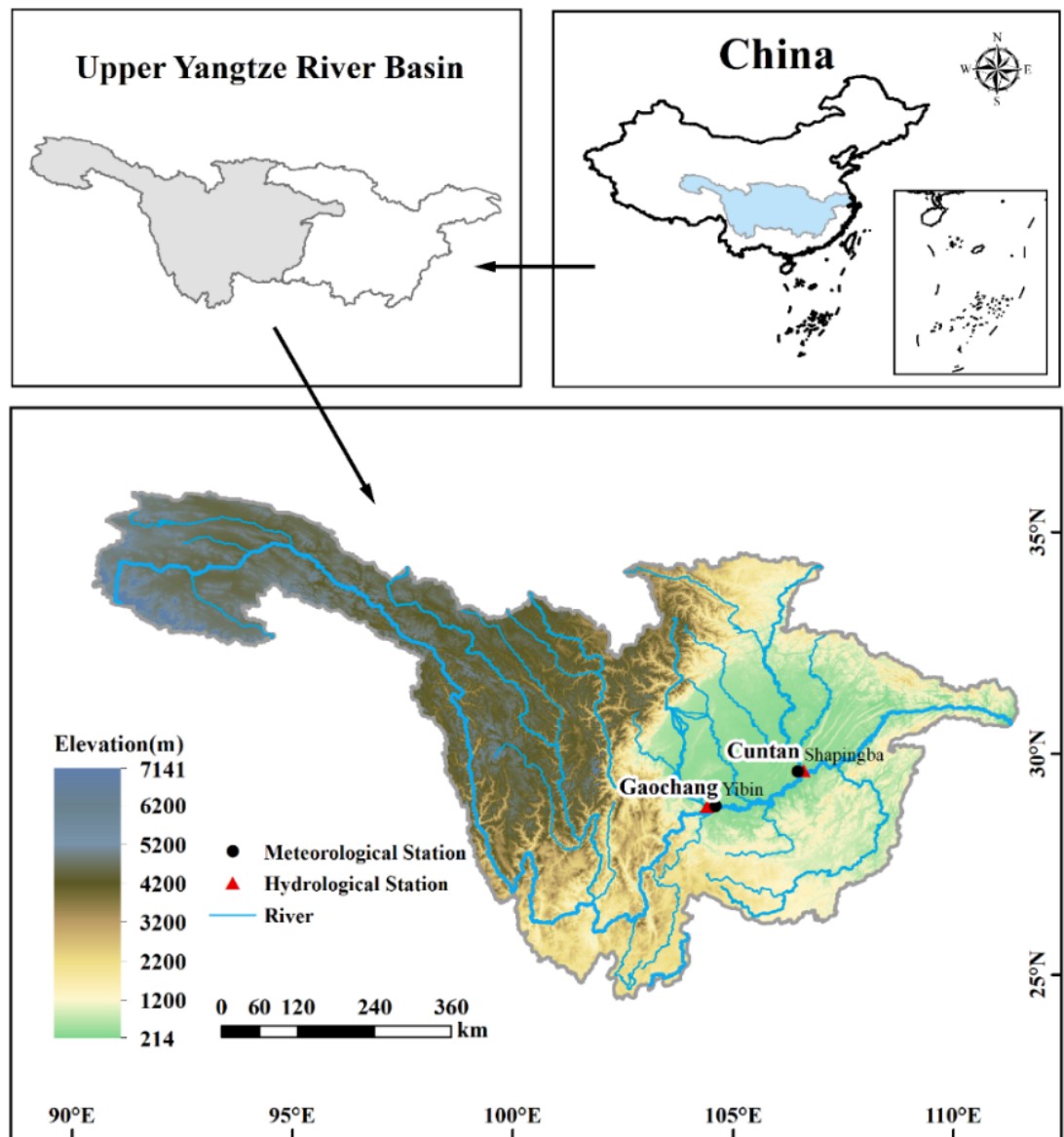

**Figure 1.** The Upper Yangtze River Basin (UYRB). This study is focused on two hydrological stations—Gaochang and Cuntan, and two meteorological stations nearby—Yibin and Shapingba.

The trend, change point, and periodicity of the monthly streamflow in the Gaochang and Cuntan stations were detected (Figure 2). No statistically significant trend (at a 5% significance level) was observed by the non-parametric Mann–Kendall test [42] and Sen's slope [43], with a *p*-value of 0.38 at Gaochang Station and 0.28 at Cuntan Station. The most probable change point was found to be NO. 593 (May 2010) at Gaochang Station and NO. 448 (April 1998) at Cuntan Station by Pettitt's Test [44], but the results did not indicate statistical significance. The Morlet wave analysis method [45] was used to analyze the periodic characteristics of the annual average and monthly runoff series. The 10–13 yearsoscillation period was most notable, the 3–5 years period was relatively notable at Gaochang Station, and the principal period and the second period of monthly runoff was found to be seven months and twelve months. While at the Cuntan Station, the 7–10 years oscillation period was most notable, and the principal period was found to be seven months with the maximum value of wavelet variance.

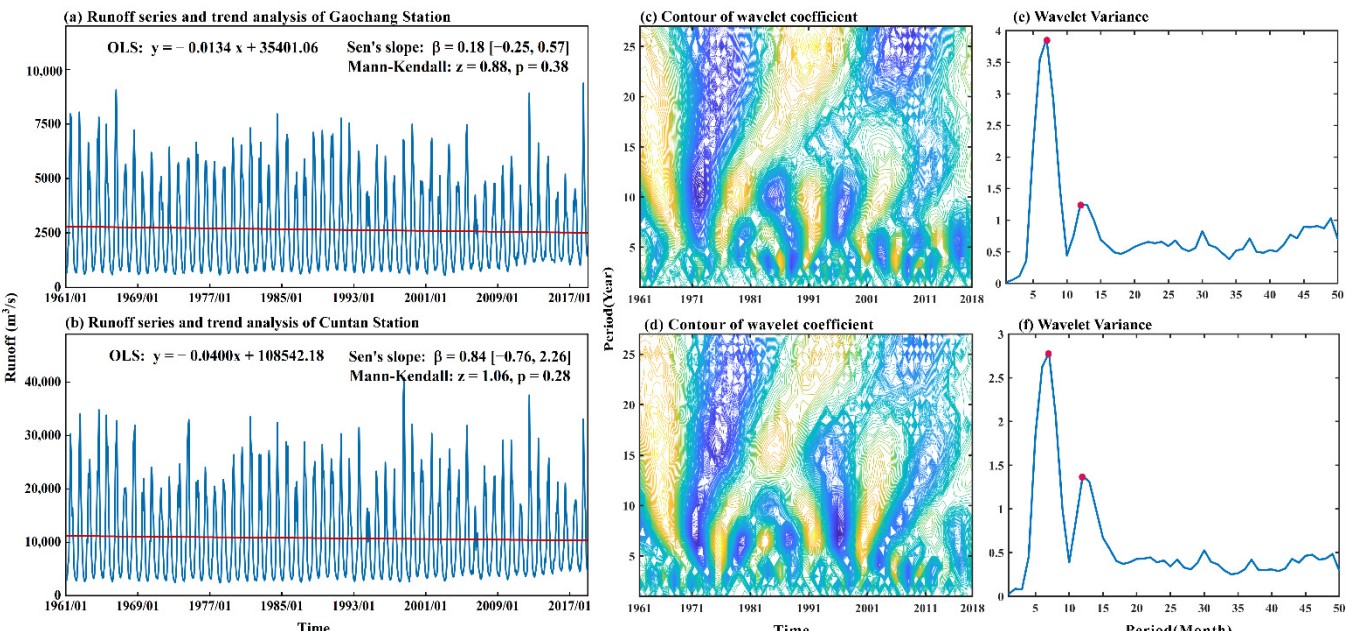

**Figure 2.** (**a**) Runoff series and trend analysis of the Gaochang Station: the blue solid line is the runoff series and the red line shows the trend of the series. The contour of the wavelet coefficient is displayed in (**c**) and the wavelet variance is shown in (**e**). (**b**,**d**,**f**) represent the same information as (**a**,**c**,**e**), but for the Cuntan Station.

Based on the analysis above, the data series was divided into two periods, i.e., January 1961–December 2008 for model training and January 2009–December 2018 for testing. Owing to the lag effect of climate causes on streamflow and the periodicity of the monthly runoff in Gaochang and Cuntan stations, 1–12 months were considered as the different lag periods when selecting the input factors in this study. Normalization of the climate and runoff data in training and testing periods was applied to refrain from the numerical problem.

### *2.2. Variable Selection*

The Pearson's correlation coefficient, stepwise regression, and copula entropy methods were applied to select input variables with different time lags. For example, to predict the runoff in January 2010, data from the previous 1 to 12 months were selected as forecast variables, and Pearson's correlation coefficient analysis first identified the top 150 variables; then the multiple stepwise regression analysis or copula entropy analysis was carried out to select the most relevant variables with the significance test.

#### 2.2.1. Pearson's Correlation Coefficient

Pearson's correlation coefficient assesses the relationship between monthly runoff and explanatory variables. The coefficient $R$ is estimated by [46]:

$$R = \frac{\sum\limits_{i=1}^{n} (x_i - \overline{x})(y_i - \overline{y})}{\sqrt{\sum\limits_{i=1}^{n} (x_i - \overline{x})^2} \sqrt{\sum\limits_{i=1}^{n} (y_i - \overline{y})^2}} \tag{1}$$

#### 2.2.2. Stepwise Regression

Stepwise regression is a multivariate regression analysis method that plays an integral role in hydrological research and modeling. The multiple stepwise regression adds the most significant variables one by one [11]. After adding a new variable at each step, an F-test is

performed to determine whether certain variables will be removed without remarkably increasing the sum of squared residuals.

### 2.2.3. Copula Entropy

The entropy of the copula function, *CE*, was used to evaluate the dependence among variables [27]. $X_1$ and $X_2$ are random variables with marginal functions $F(x_1)$, $F(x_2)$, and $U_1 = F(x_1)$, $U_2 = F(x_2)$, $u_1$ and $u_2$ represent a particular value of $U_1$ and $U_2$. The definition of *CE* is as follows:

$$H_C(U_1, U_2) = -\int_0^1 \int_0^1 c(u_1, u_2) \log c(u_1, u_2) du_1 du_2 \tag{2}$$

where $c(u_1, u_2)$ is the copula probability density function and is identical to $\frac{\partial C(u_1, u_2)}{\partial u_1 \partial u_2}$.

Mutual information (*MI*) is denoted as [47]:

$$\begin{aligned} T(X_1, X_2) &= H(X_1) + H(X_2) - H(X_1, X_2) \\ &= -H_C(U_1, U_2) \end{aligned} \tag{3}$$

*MI* can detect nonlinear correlations between target input and output [48]. However, this method cannot handle the redundant inputs directly [49], and to deal with this issue, Sharma [47] proposed partial mutual information (*PMI*). The *PMI* can be derived with the *CE* method:

$$\begin{aligned} PMI &= \int \int f_{X'Y'}(x', y') \ln\left[\frac{f_{X'Y'}(x', y')}{f_X(x')f_{Y'}(y')}\right] dx' dy' \\ &= -H_C(x', y') \end{aligned} \tag{4}$$

Fernando and May [50] suggested the Hampel test as the termination criteria:

$$Z_i = \frac{d_i}{1.4826 d_i^{(50)}} \cdot d_i = \left| CE_i - CE^{(50)} \right| \tag{5}$$

where: $Z_i$—Hampel distance; 1.4826—normalization variables; $d_i^{(50)}$—median of $d_i$; $CE_i$—the copula entropy of the $i$ th variables; $CE^{(50)}$—median *CE* values for variables set; Based on the 3σ principle, when the Hampel distance is above 3, add the candidates to the input.

In this paper, the R (R Core Team 2020) package 'copent' (https://github.com/majianthu/copent (accessed on 20 June 2022)) was applied, which implements the non-parametric method for estimating *CE* [27].

### 2.3. Prediction Models

The application of LSTM, GRU, GDBT, and RF relies on Python 3.9 with the "Scikit-Learn" package, and the libsvm package in MATLAB R2020 (a) (https://www.csie.ntu.edu.tw/~cjlin/libsvm/index.html (accessed on 20 June 2022)) was employed in the SVR prediction.

### 2.3.1. Long Short-Term Memory (LSTM)

The LSTM model comprises the input, hidden, recurrent, and output layers [48]. The memory block in the recurrent layer facilitates the interaction between the three layers, which involves multiple memory cells and three multiplier units [49]. The fundamental construction of an LSTM memory cell is displayed in Figure 3a. These three gates serve as filters [50]: The forget gate determines what message will be excluded, the input gate sets what new message will be collected, and the output gate specifies the output message from the cell condition [51].

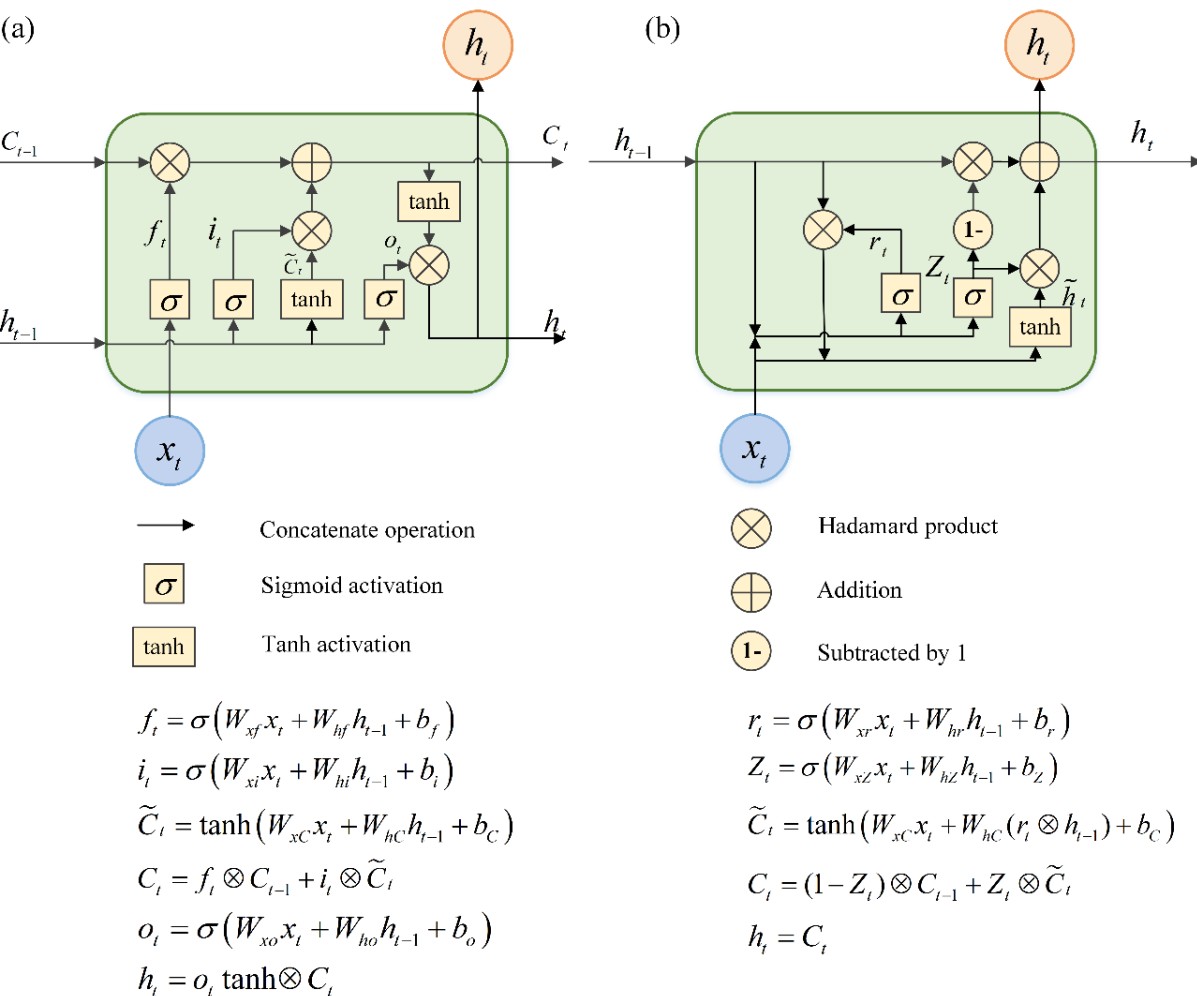

$$f_t = \sigma\left(W_{xf}x_t + W_{hf}h_{t-1} + b_f\right)$$
$$i_t = \sigma\left(W_{xi}x_t + W_{hi}h_{t-1} + b_i\right)$$
$$\widetilde{C}_t = \tanh\left(W_{xC}x_t + W_{hC}h_{t-1} + b_C\right)$$
$$C_t = f_t \otimes C_{t-1} + i_t \otimes \widetilde{C}_t$$
$$o_t = \sigma\left(W_{xo}x_t + W_{ho}h_{t-1} + b_o\right)$$
$$h_t = o_t \tanh \otimes C_t$$

$$r_t = \sigma\left(W_{xr}x_t + W_{hr}h_{t-1} + b_r\right)$$
$$Z_t = \sigma\left(W_{xZ}x_t + W_{hZ}h_{t-1} + b_Z\right)$$
$$\widetilde{C}_t = \tanh\left(W_{xC}x_t + W_{hC}(r_t \otimes h_{t-1}) + b_C\right)$$
$$C_t = (1 - Z_t) \otimes C_{t-1} + Z_t \otimes \widetilde{C}_t$$
$$h_t = C_t$$

**Figure 3.** (**a**) The construction of a fundamental LSTM cell. (**b**) The construction of a fundamental GRU cell. In Figure 3a, $W_{xi}$, $W_{hi}$, $W_{xf}$, $W_{hf}$, $W_{xo}$, $W_{ho}$, $W_{xC}$, and $W_{hC}$ are the network weights matrices. $b_i$, $b_f$, $b_o$, and $b_c$ are bias vectors. $f_t$, $i_t$, and $o_t$ are the activation value vectors of the forget gate, the input gate, and the output gate. Similarly, in Figure 3b, $W_{xr}$, $W_{hr}$, $W_{xZ}$, $W_{hZ}$ are the network weights metrics, $b_r$ and $b_Z$ are bias vectors. $r_t$ and $Z_t$ are vectors for the update and reset gate activation values.

### 2.3.2. Gate Recurrent Unit (GRU)

GRU networks were proposed to modify LSTM networks [52] with a simpler structure and faster speed. The fundamental construction of the GRU cell is displayed in Figure 3b. The hidden state ($h_t$) and cell state ($C_t$) are merged in the GRU. There are two control gates in the cell: the update gate ($Z_t$) and the reset gate ($r_t$) [53]. The update gate controls the extent to which the message from the prior step $t-1$ will be passed to the current step $t$ [48]. The reset gate determines how much information of the prior state is written into the current candidate set $\widetilde{C}_t$.

### 2.3.3. Gradient Boosted Decision Tree (GBDT)

A GBDT regression model is constructed using various decision trees (DTs) [54]. In every iteration, the latest DT is trained according to the residuals of the prior DTs based on the negative gradient, which has been confirmed to be an efficient, precise, low-bias algorithm [18,55]. This study uses the Gaussian distribution as a loss function to minimize the squared error.

### 2.3.4. Random Forest (RF)

RF is an ensemble of the decision tree model based on the bagging method [54]. As a white-box model, RF samples the raw data and generates many training samples by bootstrapping. The bagging method may address the overfitting problem of forecasting models [20,56].

### 2.3.5. Support Vector Regression (SVR)

Support vector regression (SVR) is employed in support vector machine (SVM) for regression tasks [57,58]. Based on Lagrange binary theorem [22], the SVR model tacitly transforms the original, low-dimension input into high-dimension space. In our research, the nonlinear radial basis function (RBF) was applied as the kernel function since it has demonstrated exemplary performance in predicting nonlinear runoff data for SVR [59].

### *2.4. Metrics of Performance Evaluation*

Four metrics were applied to evaluate the performance of the forecasting models, including the mean absolute percentage error (*MAPE*), root mean square error (*RMSE*), Nash–Sutcliffe efficiency coefficient (*NSE*), and Pearson's correlation coefficient (*R*), which are specified as follows:

$$MAPE = \frac{1}{n} \times \sum_{i=1}^{n} \left| \frac{Y_i - \hat{Y}_i}{Y_i} \right| \times 100\% \tag{6}$$

$$RMSE = \sqrt{\frac{1}{n} \times \sum_{i=1}^{n} \left( \hat{Y}_i - Y_i \right)^2} \tag{7}$$

$$NSE = 1 - \frac{\sum_{i=1}^{n} \left( Y_i - \hat{Y}_i \right)^2}{\sum_{i=1}^{n} \left( Y_i - Y_{avg} \right)^2} \tag{8}$$

$$R = \frac{\sum_{i=1}^{n} \left[ \left( Y_i - Y_{avg} \right) \left( \hat{Y}_i - \hat{Y}_{avg} \right) \right]}{\sqrt{\sum_{i=1}^{n} \left( Y_i - Y_{avg} \right)^2} \sqrt{\sum_{i=1}^{n} \left( \hat{Y}_i - \hat{Y}_{avg} \right)^2}} \tag{9}$$

where $n$ is the amount of data, $Y_i$ and $\hat{Y}_i$ represent the $i$th observation and prediction, and $Y_{avg}$ and $\hat{Y}_{avg}$ are the average of all the observation and prediction.

*MAPE* and *RMSE* can evaluate models' performance, especially high streamflows; *NSE* is used to test the deviation of a forecasting model, ranging from $-\infty$ to 1 [60]; *R* is adapted to evaluate the linear correlation between observation and prediction. Models with larger *R* and *NSE* values or smaller *MAPE* and *RMSE* values indicate better predictive performance [61].

### *2.5. Model Calculation Scheme*

Cause-driven multivariate forecasting models reflect the relationship between variables and forecast elements regarding runoff causation [18]. The selection of variables and forecast models are crucial when applying this approach. The model calculation scheme is shown below (Figure 4).

First, the essential variables were chosen as the model input set. The candidate predictive variables include one hundred thirty global circulation indexes, eight surface meteorological data, and the antecedent runoff (Table S1). The total number of variables is $139 \times 12$, considering the lag time of 1–12 months. Subsequently, the selected variables by different methods were input into five ML models, and four classical univariate time series models were used as benchmarks. Last, four indicators were used to measure the performance of the prediction models. Thus, the optimal monthly runoff prediction model and a few sub-optimal models were recommended by comparing a weighted average score of four metrics for a specific station.

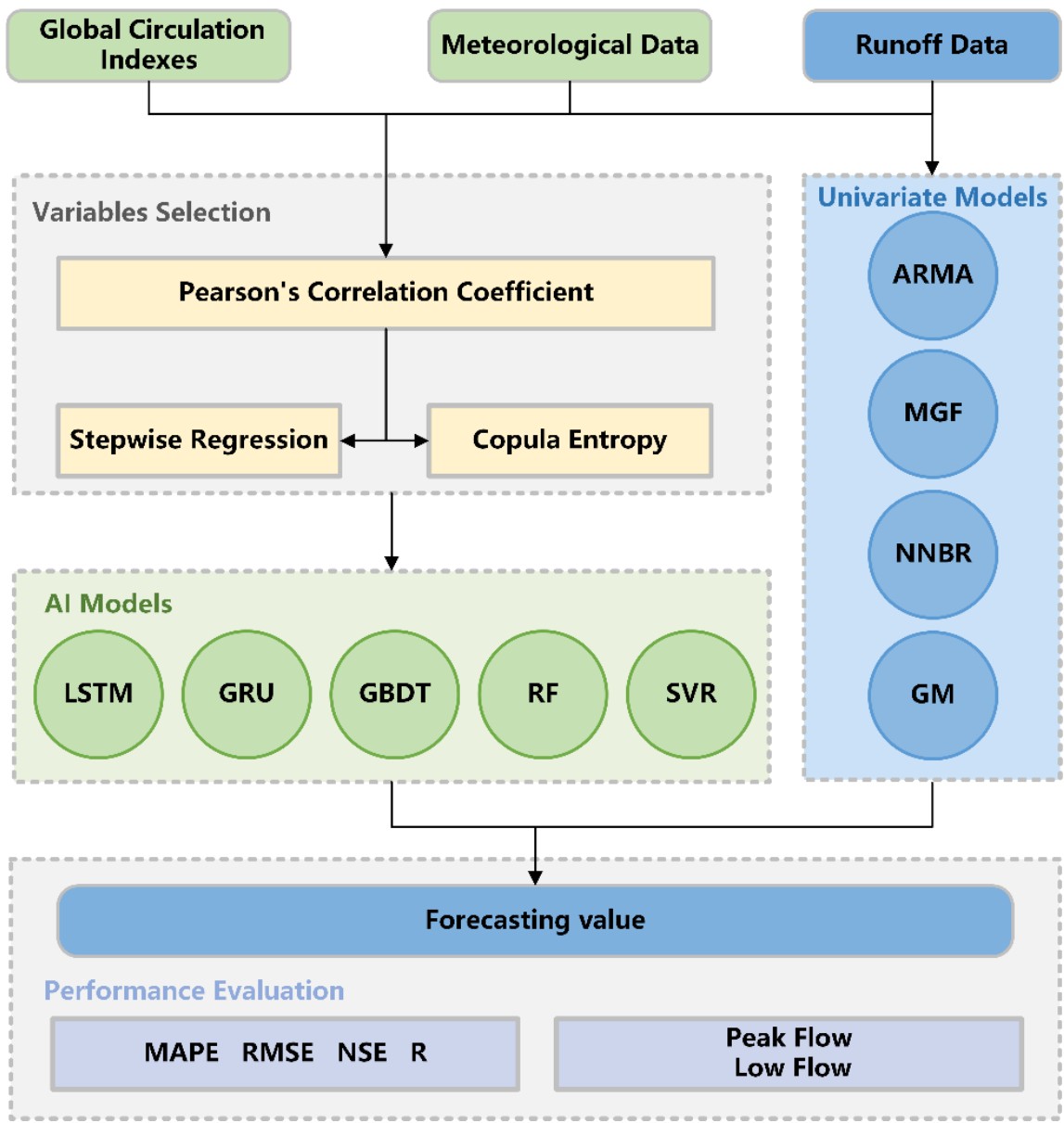

**Figure 4.** Flowchart of model development for predicting monthly runoff.

## 3. Results

### 3.1. Variable Selection

The top 150 variables were first identified by Pearson's correlation coefficient analysis, followed by the multiple stepwise regression analysis or copula entropy analysis to select the most relevant variables. The calculated copula entropy and $Z$ values of the first 150 variables are shown in Figure 5. $Z$ represents the $CE$ value after the Hample test, which fluctuates more than $CE$ at both the Gaochang Station and the Cuntan Station. Based on the $3\sigma$ criterion, when $Z$ is greater than 3, the candidate variable has a significant entropy value and can be added to the input set. However, none of the variables had a $Z$ value greater than 3. Hence, a lower confidence level was considered in this paper and the top ten variables in terms of $Z$ value were identified as an input set for the forecasting models. To keep the number of variables in the input set consistent, the top ten variables were selected in stepwise regression analysis after the multicollinearity testing and correction.

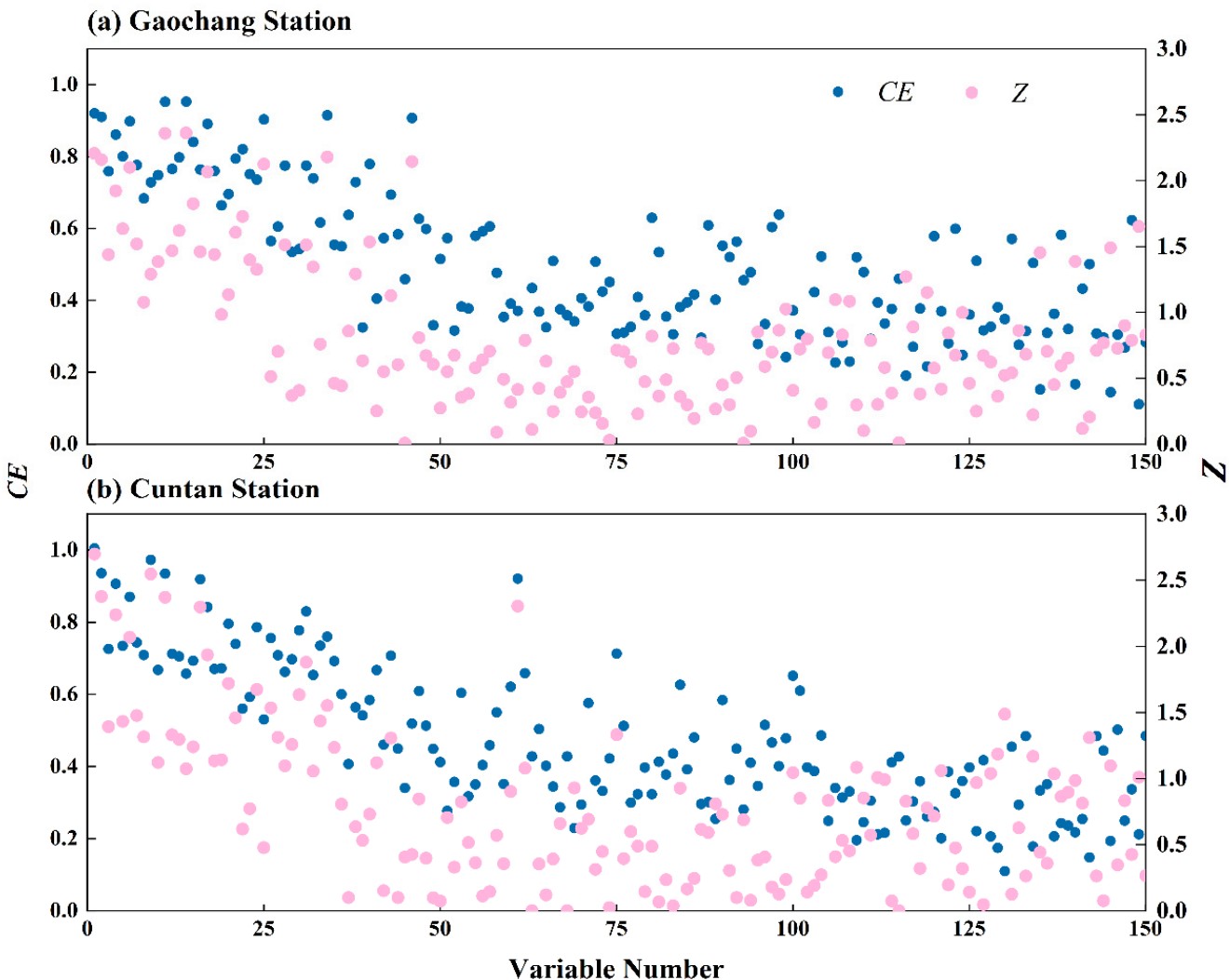

**Figure 5.** Variable selection results for the Gaochang Station (**a**) and Cuntan Station (**b**) by the copula entropy method. *CE* denotes the copula entropy, and *Z* represents the Hampel distance after the Hample test. The greater the *CE* value or *Z* value, the more significant the effect of the corresponding variable.

The variables selected by stepwise regression and copula entropy are listed in Table 1. For the Gaochang Station, average temperature, runoff, and maximum temperature were selected by stepwise regression and copula entropy methods. The stepwise regression method selected more global circulation variables, such as the northern hemisphere polar vortex central intensity index, Tibet Plateau region 1 index, Asia polar vortex area index, and so on; while the copula entropy method selected more ground meteorological variables. Most of the variable's lag was 1 month, followed by 12 months and 6 months, which is synchronous with the natural period of the water cycle. For the Cuntan Station, the maximum temperature was the most significant variable selected both by the stepwise regression method and the copula entropy method. Besides, runoff with 1 month, 6 months, and 12 months' time lag were selected, indicating a strong autocorrelation in streamflow. Likewise, the tepwise regression tended to select more global circulation variables than the Copula Entropy method at the Cuntan Station.

**Table 1.** Variables selected by stepwise regression and copula entropy for the Gaochang Station and Cuntan Station.

| Station | Stepwise Regression | | Copula Entropy | |
|---|---|---|---|---|
| | Variables | Lag (Month) | Variables | Lag (Month) |
| Gaochang | Average Temperature | 7 | Maximum Temperature | 1 |
| | Runoff | 12 | East Asian Trough Intensity Index | 6 |
| | Northern Hemisphere Polar Vortex Central Intensity Index | 1 | Average Temperature | 7 |
| | Maximum Temperature | 6 | Daylight Hours | 2 |
| | North American Subtropical High Area Index | 12 | Maximum Temperature | 7 |
| | Runoff | 1 | Runoff | 6 |
| | Relative Humidity | 1 | Daylight Hours | 1 |
| | Tibet Plateau Region 1 Index | 5 | East Asian Trough Intensity Index | 12 |
| | Asia Polar Vortex Area Index | 1 | Average Temperature | 1 |
| | Indian Ocean Warm Pool Strength Index | 9 | Runoff | 12 |
| Cuntan | Maximum Temperature | 7 | Maximum Temperature | 7 |
| | Runoff | 12 | Maximum Temperature | 1 |
| | Northern Hemisphere Polar Vortex Intensity Index | 2 | Average Temperature | 7 |
| | Runoff | 1 | East Asian Trough Intensity Index | 7 |
| | North American Subtropical High Intensity Index | 12 | Runoff | 6 |
| | Atlantic-European Polar Vortex Intensity Index | 7 | Average Temperature | 1 |
| | Daylight Hours | 12 | Runoff | 12 |
| | Asia Polar Vortex Intensity Index | 6 | East Asian Trough Intensity Index | 1 |
| | Eurasian Zonal Circulation Index | 9 | Daylight Hours | 8 |
| | Air Pressure | 3 | Daylight Hours | 2 |

*3.2. Model Structure and Parameter Selection*

In the development of LSTM and GRU, the num_layers and batch_size were set to 2 and 12, separately, and the hidden_size from 60 to 120 was examined to identify the optimal networks. In addition, the epochs were set as 1000 times, and the learning rate was set as 0.0005, while the best models were evaluated by minimum *MARE* in the testing stage. For the Gaochang Station, the optimal parameters (hidden_size and epoches) of the LSTM_Step model, LSTM_Copula model, GRU_Step model, and GRU_Copula model were (112, 862), (94, 977), (68, 562), and (75, 788) respectively. For the Cuntan Station, the optimal parameters of the LSTM_Step model, LSTM_Copula model, GRU_Step model, and GRU_Copula model were (70, 357), (95, 488), (85, 273), and (106, 634). The parameter optimization process of hidden_size in the LSTM_Copula model at Gaochang Station was demonstrated in Figure 6a, where the model performance improved when the hidden_size was in a range of 90~100 but decreased when the hidden_size was more than 100. The performance of the GRU_Step model at the Cuntan Station was improved as the epochs increased but reduced when ephochs were larger than 330 (Figure 6b).

In the modeling with GDBT and RF, the hyperparameter setting is a key step, where randomized search and grid search were used sequentially. Firstly, the n_estimators (the number of decision trees), max_depth (the maximum depth of decision trees), min_samples_split (the minimum number of split samples), min_samples_leaf (the minimum sample size of the nodes), and max_features (the maximum sampling ratio) in the hyperparameter were set in a relatively wide range, and 200 random iterations were performed with 3-fold cross-validation using RandomizedSearchCV.py. Based on the result of the randomized search, a few values were selected in the nearby range, and each match was traversed through GridSearchCV.py to search for the optimal hyperpa-

rameter values (Figure 6c). For the Gaochang Station, the optimal hyperparameters of the GDBT_Step model, GDBT_Copula model, RF_Step model, and RF_Copula model are (230, 38, 9, 1, 2), (910, 50, 3, 2, 3), (380, 14, 11, 2, 4), (127, 28, 3, 2, 1) respectively. For the Cuntan Station, the optimal hyperparameters of the GDBT_Step model, GDBT_Copula model, RF_Step model, and RF_Copula model are (50, 35, 6, 1, 2), (100, 22, 11, 1, 2), (90, 46, 11, 2, 4), (270, 17, 2, 1, 3) respectively.

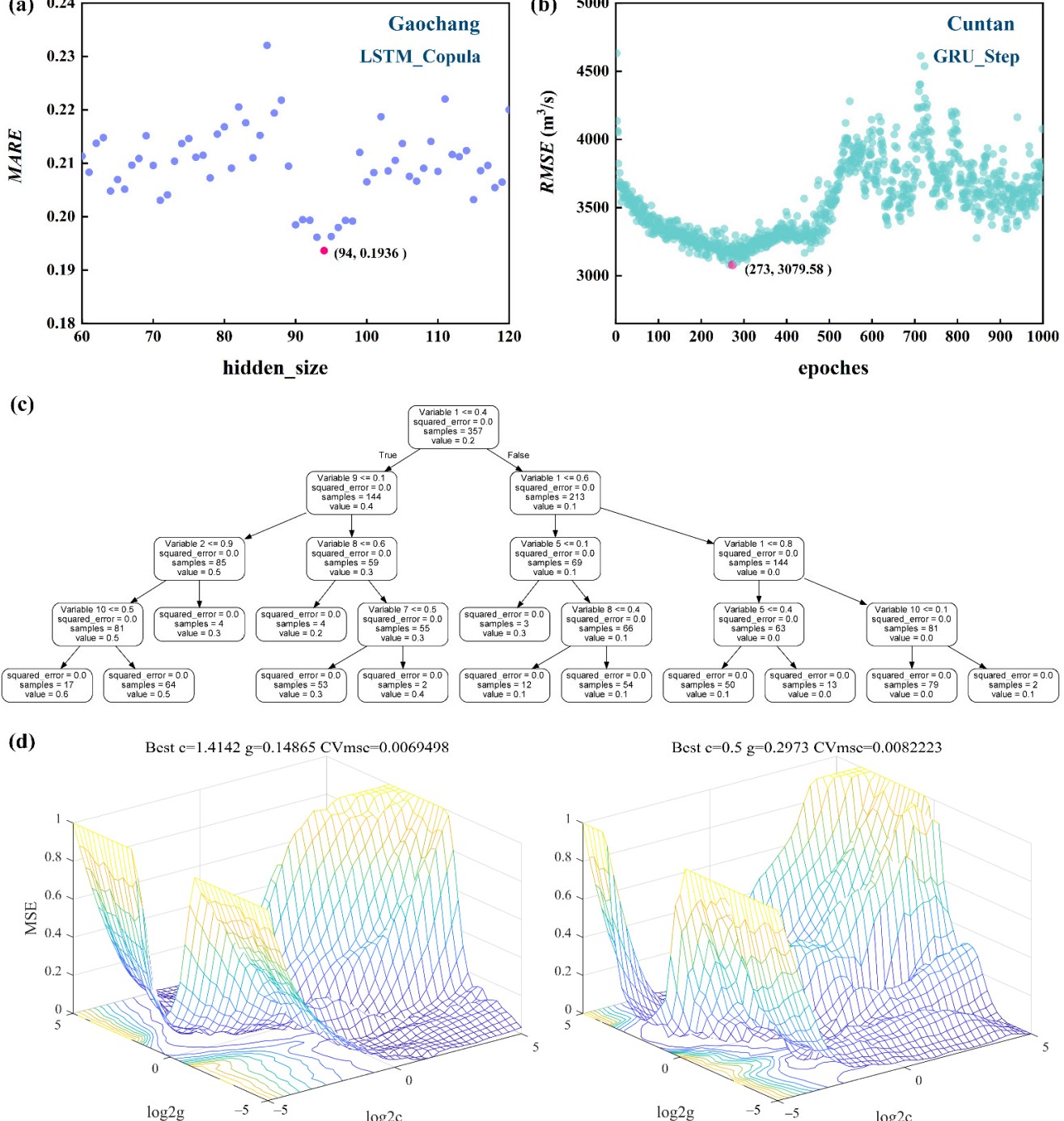

**Figure 6.** (**a**) The parameter optimization process of hidden_size in the LSTM_Copula model at Gaochang Station. (**b**) The optimal result of epochs in the GRU_Step model at Cuntan Station. (**c**) The simplified decision tree visualizing plot in the RF_Step model at Cuntan Station. (**d**) The best cost (c) and gamma (g) in the SVR_Step and SVR_Copula model at Gaochang Station.

The grid search method of 3-fold cross-validation is applied to find the best values for cost (c) and gamma (g), two crucial parameters in the SVR model (Figure 6d). For the Gaochang Station, the optimal parameters of the SVR_Step model and SVR_Copula model are (1.41, 0.15), (0.50, 0.29), respectively. For the Cuntan Station, the optimal parameters of the SVR_Step model and SVR_Copula model are (3.0, 0.29), (6.0, 0.42) respectively.

### 3.3. Comparison of Various Models' Performance

Among the ten models examined herein, the GBDT_Step and GBDT_Copula models achieved the best *MAPE*, *RMSE*, *NSE*, and *R* values in the training stage for the two stations (Table S2). In the testing period, the GRU_Copula model obtains the best *MAPE* values of 16.68%, and the LSTM_Copula model performs better than other models in *RMSE*, *NSE*, and *R* for the Gaochang Station. While for the Cuntan Station, the SVM_Step model obtains the best *MAPE* values of 13.98%, followed by LSTM_Copula of 14.44%; the RF_Copula model achieves the best *RMSE* values of 2616 $m^3/s$, and LSTM_Copula accomplishes the best *NSE* and *R* values.

The copula entropy method had a more consistent performance effect across the months than the stepwise regression method, which was distinct in LSTM, GBDT, and SVR models (Figure S1). For the Gaochang Station, the LSTM_Copula model outperformed other models with better values among four evaluation metrics and a more robust effect on different months. For the Cuntan Station, models performed differently on different indicators, but LSTM_Copula, GRU_Step, and RF_Copula models performed better overall.

All the ML models could track the observed changes in the runoff series, manifesting the validity of the ML models (Figures 7 and 8). For the Gaochang Station, GBDT and SVM models were superior to the other methods because the deviations between the observations and simulations were small, and the high flows above 6000 $m^3/s$ were better captured. In addition, the two selection methods did not show much difference when applied in the five ML models. For the Cuntan Station, GBDT and RF models outperformed other models from the denser results in the scatter plots; and the Copula method performed better in LSTM and SVR models. Given that the peak flows at two stations were not well predicted, the ten models need to be improved in simulating hydrological extremes.

To further illustrate the advantages of using multi-variable inputs for data-driven models, four classical univariate time series models were used as benchmarks: ARMA, MGF, NNBR, and GM (Table S3 and Figure S2). In the testing stage, the ARMA model outperformed other models with the best *MAPE*, *RMSE*, and *NSE* values for the Gaochang Station. In addition, the ARMA model had a more concentrated distribution of monthly average *MAPE* and *NSE,* while the MGF model outperformed on *RMSE* and *R*. For the Cuntan Station, the ARMA model achieved the best *MAPE* values of 14.39% and the best NSE of 0.84, while the MGF model outperforms the ARMA with an *R*-value of 0.96. Similarly, the ARMA model's *MAPE* and *NSE* have more concentrated distributions than other univariate models, and the MGF model outperforms on *RMSE* and *R*. The same results can be found in univariate models where the observed and predicted values overlap and cluster.

Overall, the four univariate models did not predict the monthly streamflow well enough like the ML models. For the Gaochang Station, the univariate models' *MAPE* ranged from 17.47~27.79%, the *RMSE* ranged from 715 to 887 $m^3/s$, and the *NSE* ranged from 0.52~0.77, except that the evaluation metric *R* was relatively better with a range of 0.89~0.95, whereas the ML model's *MAPE* ranged from 16.68~26.26%, the *RMSE* ranged from 691~844 $m^3/s$, the *NSE* fluctuated from 0.58 to 0.78, the *R* fluctuated from 0.91~0.94, which illustrated the improvement of the ML model over the univariate model in terms of *MAPE* by 5.10, *RMSE* by 4.16, *NSE* by 5.34, and *R* by 0.43%. The evaluation of the four models was quite the same for the Cuntan Station, and the improvement of the ML model over the univariate model was more apparent, with *MAPE* by 10.84, *RMSE* by 17.28, *NSE* by 13.68, and *R* by 3.55%.

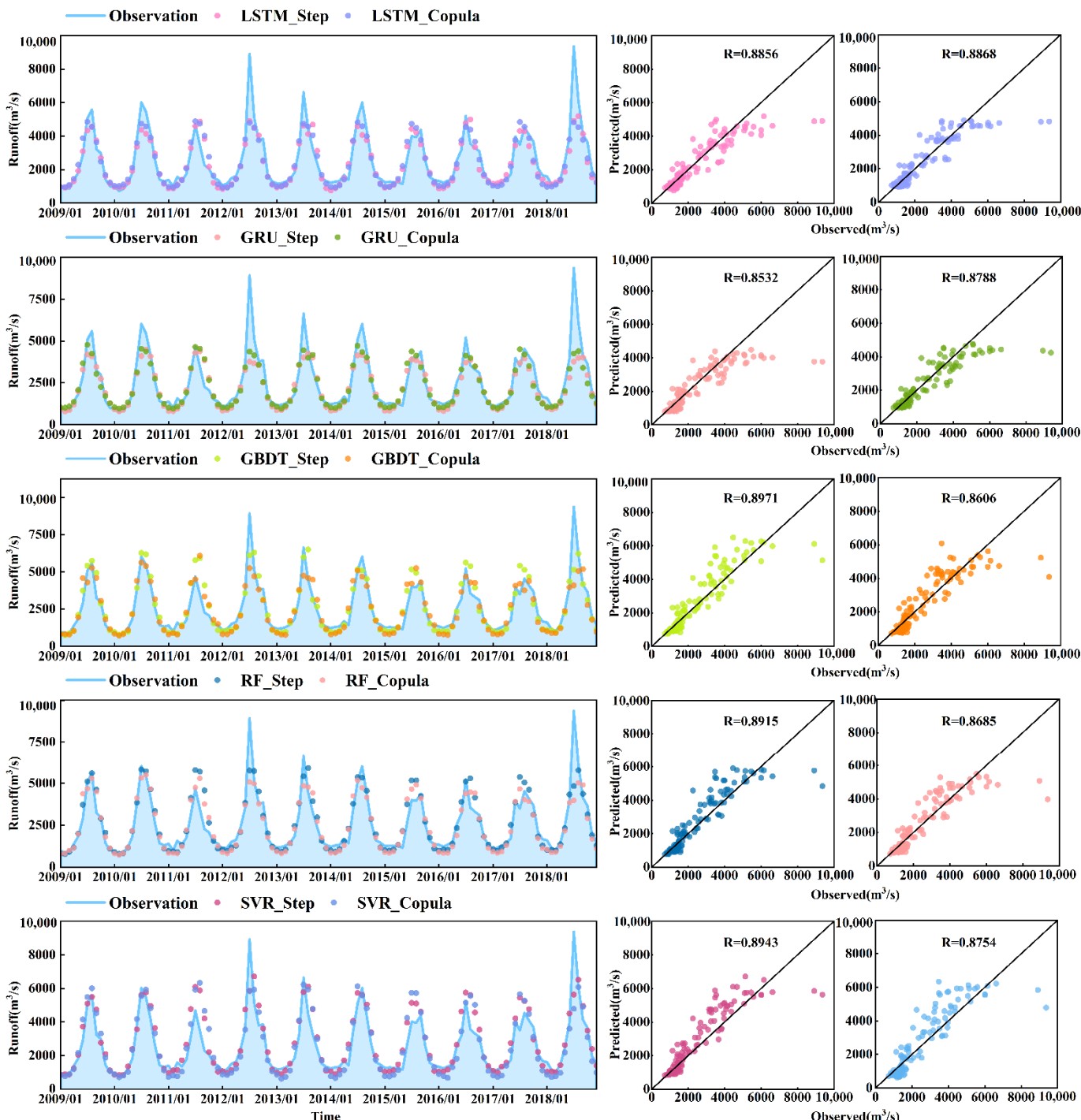

**Figure 7.** Comparison of simulated with observed monthly runoff in the testing stage by the machine learning models for the Gaochang Station.

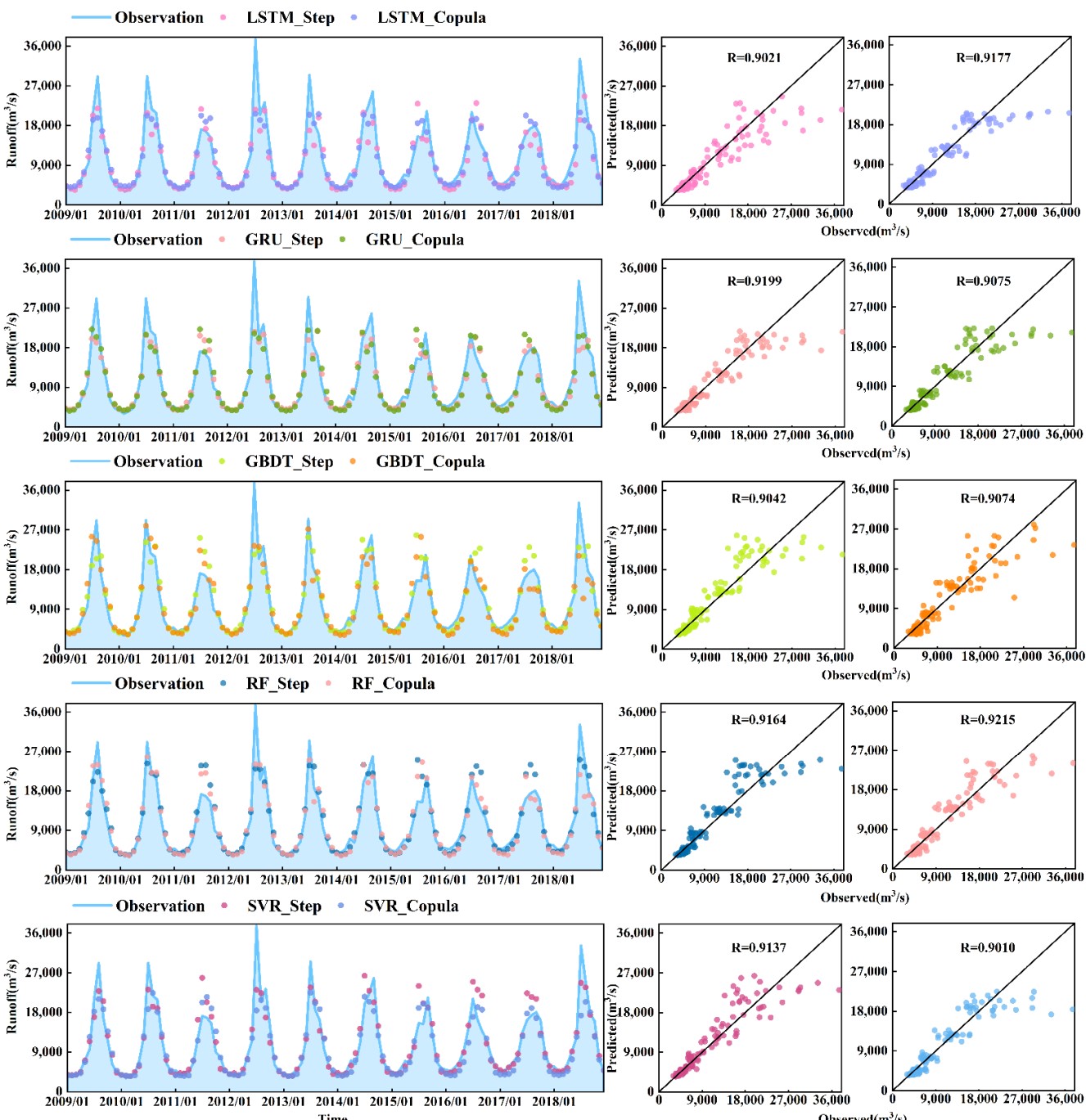

**Figure 8.** Comparison of simulated with observed monthly runoff in the testing stage by the machine learning models for the Cuntan Station.

The performance of these 14 models was quantitatively compared based on the entropy weight method (Figure 9). Four evaluation metrics, *MAPE*, *RMSE*, *NSE*, and *R*, were considered comprehensively through the index weight and normalized value calculated by the entropy method, where *MAPE* and *RMSE* were negative metrics while *R* and *NSE* were positive metrics. It turned out that the index weights were close both in the Gaochang Station and the Cuntan Station, ranging from 0.249~0.251. The best weighted average score for the Gaochang Station was 0.0780 from the LSTM_Copula model, followed by the GRU_Copula with a score of 0.0771; while the lowest score was 0.0599 in the MGF model. For the Cuntan Station, the best weighted average score was 0.0758 in the LSTM_Copula model, while the lowest score was 0.0570 in GM. The weighted average performance score

for the Gaochang Station differed significantly among these 14 models, whereas for the Cuntan Station, the ML models indicated a similar performance except for the LSTM_Step model, and there was a significant advantage compared with the univariate models, with an average improvement of 0.0093.

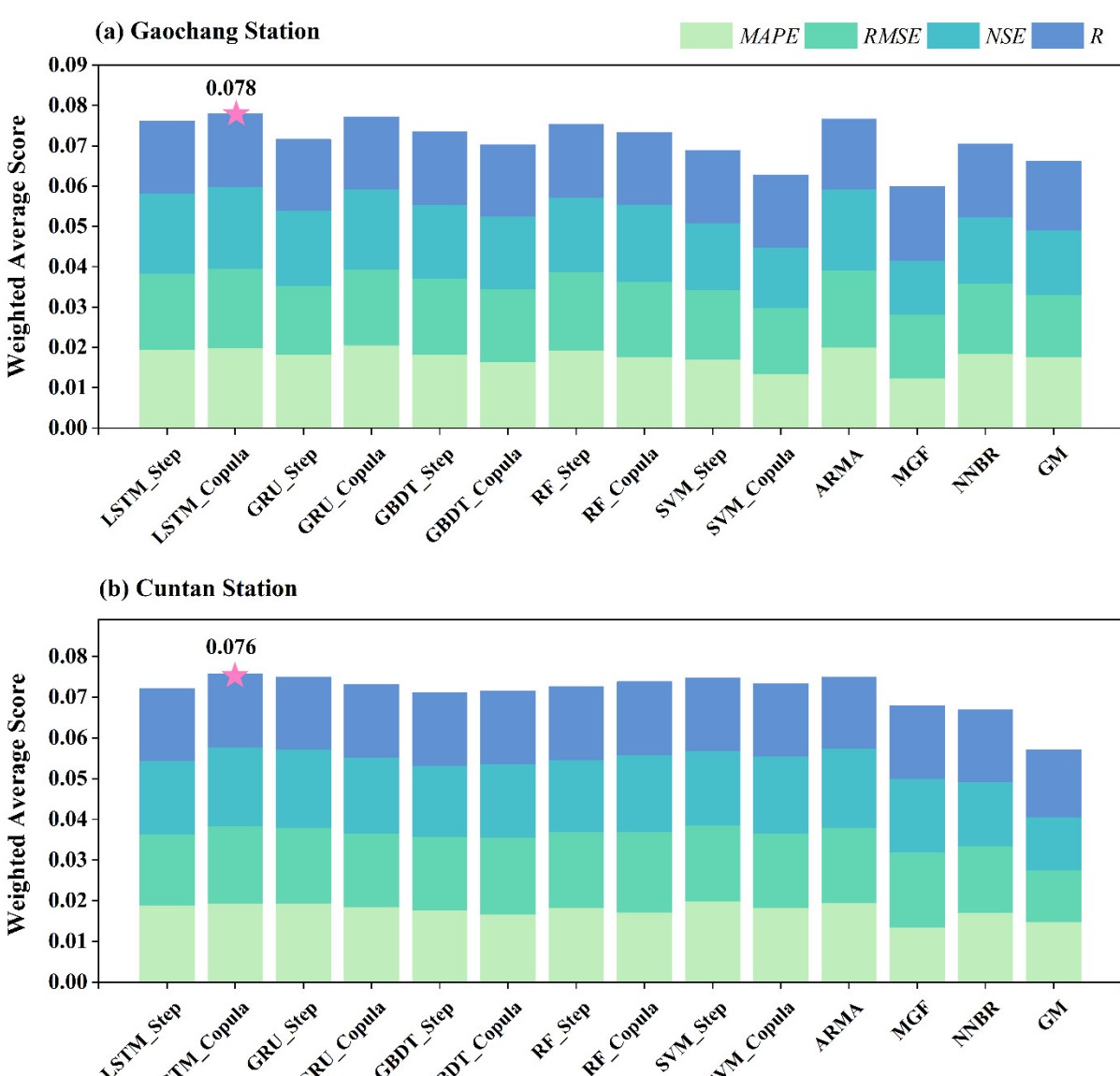

**Figure 9.** The weighted average scores of *MAPE*, *RMSE*, *NSE*, and *R* by machine learning models and univariate models at the Gaochang Station (**a**) and the Cuntan Station (**b**) in the testing stage. The pink pentagram denotes the best score.

### 3.4. Accuracy of Peak Flow and Low Flow Forecasts

Generally, the peak flows were not captured well, and all models failed to capture the most severe peak flow (Figures S3 and S4). For the Gaochang Station, the peak flows in 2012 and 2018 were not captured by any of the ten ML models or the four univariate models (Figure S3). The observed peak flow in 2012 was 8921 m$^3$/s, but the average predicted peak flow was only 5278 m$^3$/s, which was even more inaccurate in 2018, with an observed peak flow of 9366 m$^3$/s and a predicted value of 5314 m$^3$/s. In other years, LSTM and GRU models tended to underestimate the peak flows to 9.58~14.34%, while GBDT, RF, and SVR tended to overestimate the peak flows to 4.94~12.50%. For the Cuntan Station (Figure S4), the GDBT_Copula model tracked the peak flow better than others in 2009, 2010, 2013, and 2017. The peak flow in 2012 was underestimated by all the ten ML models to an average

of 41.06% and by the four univariate models to an average of 34.8%. The highest peak flow in 2018 was underestimated by the ten ML models to an average of 33.01% and by the four univariate models to an average of 29.85%. Besides, the peak flow in 2011 was overestimated by the ten ML models to an average of 31.51% and by the four univariate models to an average of 24.15%.

The predictions on the low flows were better than peak flows (Figures S3 and S4). For the Gaochang Station, generally, all ML models and univariate models tended to underestimate the low flows except the year 2010 to an average relative error of 14.49% (Figure 10). The distributions of relative error on annual peak flow prediction were denser than on annual low flows in the testing stage, especially LSTM_Copula model, GBDT_Step model, and RF_Step model. Besides, NNBR and GM did not show stable performance in predicting peak and low flows. For the Cuntan Station, the prediction performances on the low flows were better than the Gaochang Station, given that the streamflow is higher at the Cuntan Station. The ML models' prediction results were close to the observation with an average relative error of 5.76%, and the distributions of ML models were denser than traditional statistical models. While the univariate models had a worse performance with an average relative error of 14.58%, MGF and GM models did not show good performance (Figure 11), indicating that the complex physical mechanism underlying extreme floods could not be captured by simple univariate models.

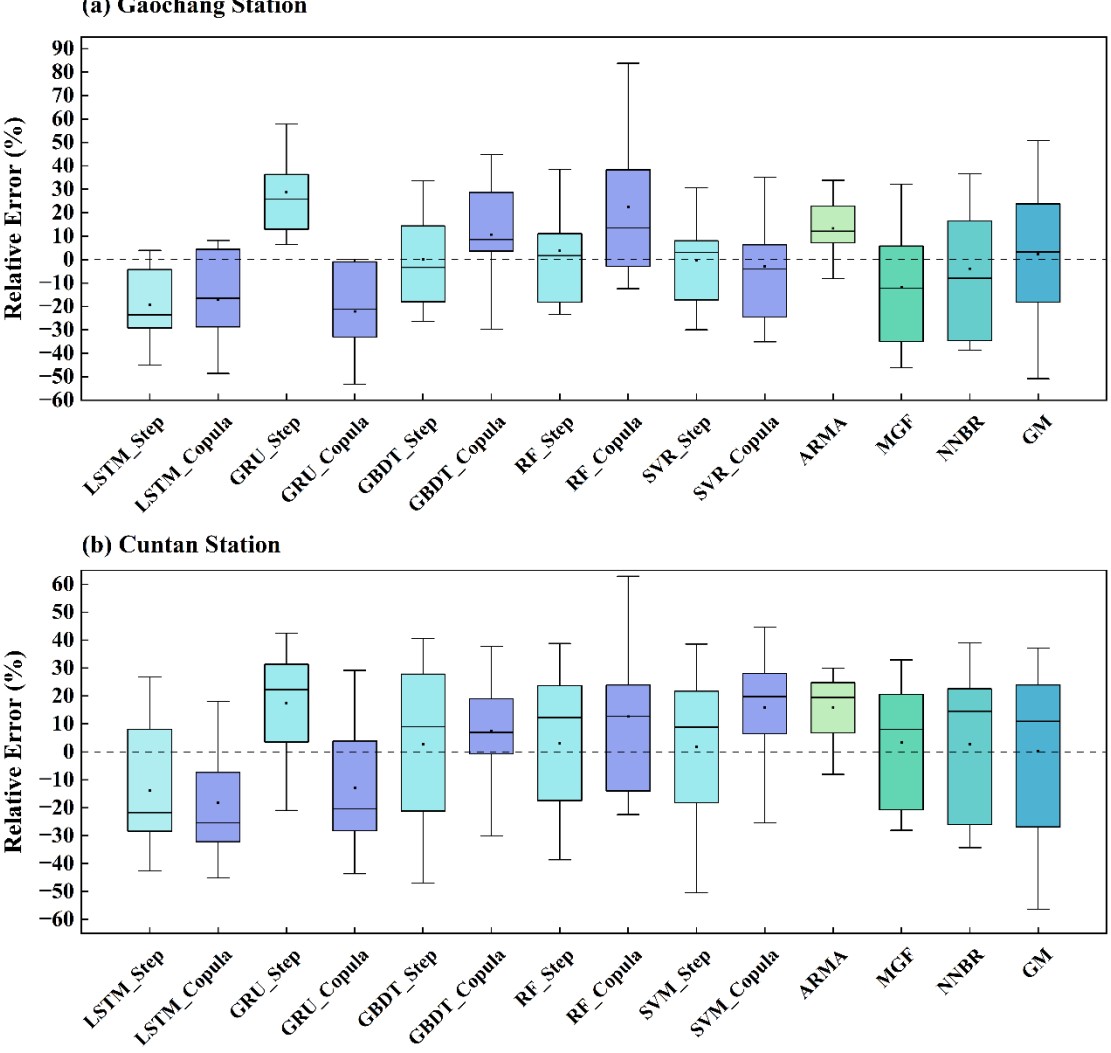

**Figure 10.** Relative errors of annual peak flow prediction by machine learning models and univariate models at the Gaochang Station (**a**) and Cuntan Station (**b**) in the testing stage.

**(a) Gaochang Station**

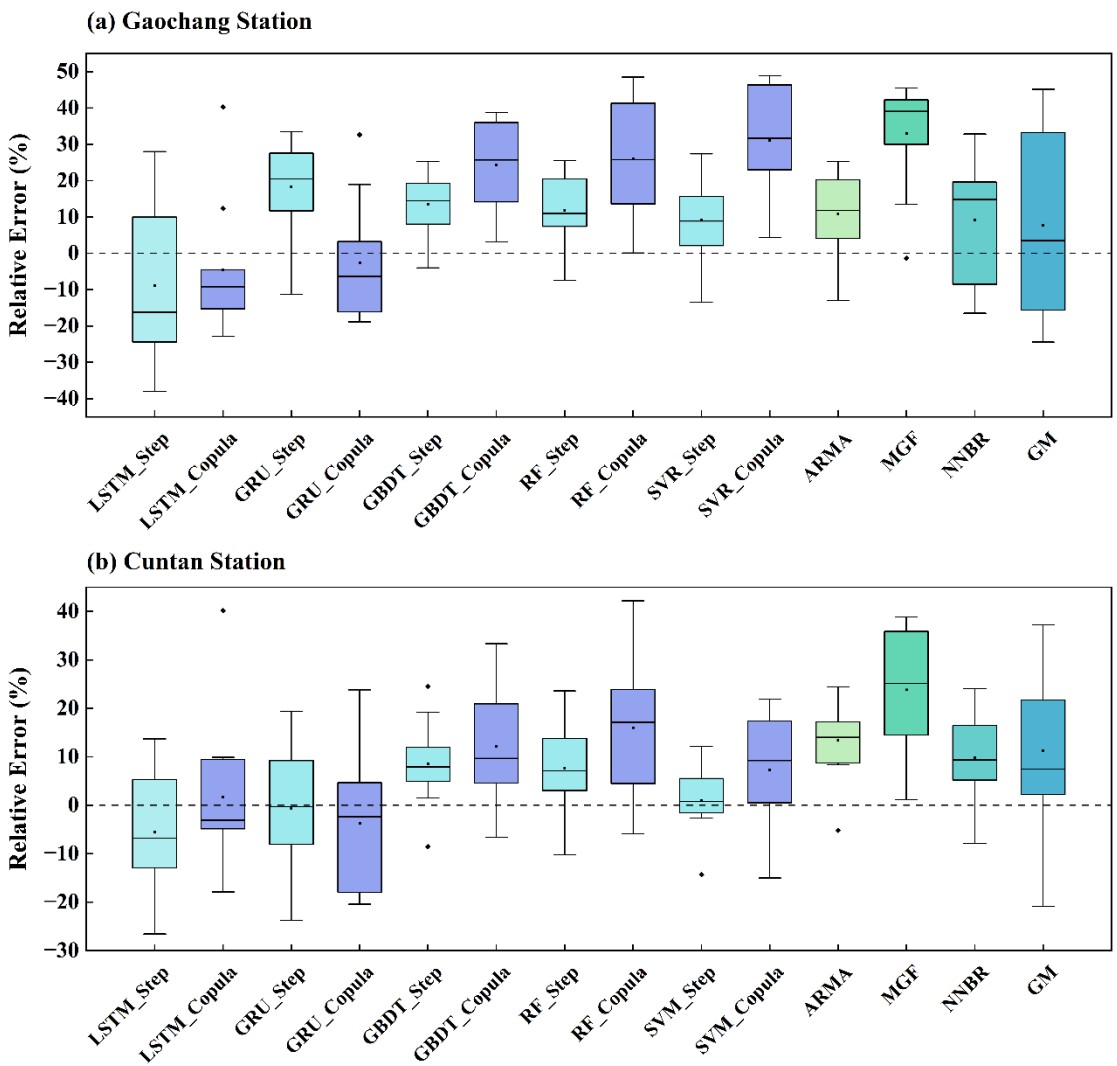

**(b) Cuntan Station**

**Figure 11.** Relative errors of annual low flow prediction by machine learning models and univariate models at the Gaochang Station (**a**) and Cuntan Station (**b**) in the testing stage. The black rhombus denote the anomaly.

## 4. Discussion

Considering the effects of different variables and lag periods on runoff, we applied five common ML models with the traditional stepwise regression method and the copula entropy method to select the optimal variables and predict monthly streamflow for the Gaochang Station and the Cuntan Station. We also applied four univariate models as benchmarks to investigate the role of multi-variable input in monthly streamflow prediction.

The results revealed that input variables of different time lag influenced the prediction performance of runoff. Interestingly, the stepwise regression method selected more global circulation variables while the copula entropy method tended to select more ground meteorological variables. At the Gaochang Station, the variables impacting the runoff process include the northern hemisphere polar vortex central intensity Index with a lag of one month, the North American subtropical high area index with a lag of twelve months, the Tibet Plateau region 1 Index with a lag of one month, the Asia polar vortex area index with a lag of one month, the Indian Ocean warm pool strength index with a lag of nine months, suggesting that the streamflow processes in the UYRB are not only closely linked geographically to the Tibetan Plateau with pre-summer thawing of frozen soil [62–65], but also remotely influenced by the atmospheric circulation in the northern hemisphere, especially the East Asian monsoon circulation system, and the warm Indian Ocean condition

and tropical SST anomalies [66–69], which can be found in the selected variables of the Cuntan Station as well.

Generally, the ML models outperform the univariate models in both the training and the testing stages. For instance, the LSTM_Copula model makes average improvements of 12.25, 8.71, 10.06, and 1.12% in the *MAPE*, *RMSE*, *NSE*, and *R* values than other ML models for the Gaochang Station and 6.59, 2.89, 5.19, and 0.66% for the Cuntan Station. In addition, the ML models outperform the univariate models on the annual low flows with much lower relative errors of 5.76 and 14.58% for the Gaochang Station and the Cuntan Station. However, the performance of ML and univariate models did not look distinct when predicting the annual peak flow, suggesting the difficulty of extreme runoff prediction. The comparative analysis above substantiates the vital part of meteorological variables input for data-driven models and confirms the superiority of the nonlinear and self-learning ML models.

The optimal models for monthly streamflow prediction differed between the two stations, which explains the complexity and difficulty of medium to long-term runoff prediction. However, some models outperformed others to a certain degree, and the comparison results illustrated the superiority of the copula entropy method and LSTM model. For the Gaochang Station, the LSTM_Copula model outperformed other models with better values of evaluation metrics and a more steady and robust effect in different months with 17.624% of *MAPE*, 691.492 m$^3$/s of *RMSE*, 0.783 of *NSE*, and 0.937 of *R* in the testing stage, and the LSTM_Copula model obtained the best weighted average score of 0.078, which was similar to the previous research [70] in the Gaochang Station with ANN, ELM, and SVM models. Compared with the LSTM_Step, the LSTM_Copula improved the *MAPE*, *RMSE*, *NSE*, and *R* values by about 3.33, 5.19, 1.68, and 1.47% in the testing stage, respectively. Besides, it increased 20.48, 10.65, 11.49, and 3.25% in the *MAPE*, *RMSE*, *NSE*, and *R* values in the testing stage in comparison with the GBDT_Copula model. For the Cuntan Station, LSTM_Copula and RF_Copula models performed relatively better in general with 14.441~16.734% of *MAPE*, 2616.354~2782.648 m$^3$/s of *RMSE*, 0.835~0.811 of *NSE*, and 0.960~0.962 of *R* in the testing stage, which is better in the *MAPE* and *R* values than the previous study [71] on the Cuntan Station.

Furthermore, some aspects limit the prediction accuracy and stability in this study. On the one hand, the ML models can theoretically reach the approximate solution, while the conventional gradient-based training technique tends to be stuck in the local minimum [72–75]. On the other hand, the calibration of hyperparameters greatly influences the forecasting models' results. Furthermore, the computational cost of the ML models is longer than traditional statistical models since the models' structure is more complex and the optimal parameter searching is time-consuming, which took 450~600 s for LSTM and GRU modeling, 120~200 s for the GBDT, RF, and SVR modeling, and 30~70 s for traditional statistical modeling in personal computers with 8 CPU and dual thread in our study. Besides, influenced by multiple variables in both the physical world and human society [76,77], the runoff process presents knotty dynamic characteristics, making it more challenging to predict. Accordingly, in some cases, a single model and limited input variables may not be able to make satisfactory predictions. Multiple-model coupling is inevitable in the future, and more work is needed to employ more input variables by novel algorithms [78–80] and to investigate the interpretability of the chosen variables based on the developed knowledge.

## 5. Conclusions

The stepwise regression and copula entropy methods were applied in five ML models for monthly streamflow prediction for the Gaochang Station and Cuntan Station in the UYRB, including the LSTM_Step, LSTM_Copula, GRU_Step, GRU_Copula, GBDT_Step, GBDT_Copula, RF_Step, RF_Copula, SVR_Step, and SVR_Copula. The results indicate that the LSTM_Copula model outperformed other models in predicting monthly runoff at the Gaochang Station and the Cuntan Station, whereas the GRU_Step and RF_Copula models also showed satisfactory performances. Besides, LSTM and GRU models tended to underestimate the peak flows while GBDT, RF, and SVR tended to overestimate them.

This means that the accuracy of peak flow forecasting still needs improvement owing to the few extreme flood samples available for learning. In addition, compared with four univariate time series models (i.e., ARMA, MGF, NNBR, and GM), the ML models with multi-variables input generally presented better forecasting accuracy. In conclusion, we demonstrate that the proposed ML methods are potentially effective tools for monthly streamflow prediction, selecting appropriate input variables and time lags simultaneously.

**Supplementary Materials:** The following supporting information can be downloaded at: https://www.mdpi.com/article/10.3390/su141811149/s1, Figure S1: Observed runoff and simulated monthly runoff by univariate models for the Gaochang Station in the testing stage; Figure S2: Observed runoff and simulated monthly runoff by univariate models for the Cuntan Station in the testing stage; Figure S3: Annual peak flows of various models for the Gaochang Station and Cuntan Station in the testing stage; Figure S4: Annual low flows of various models for the Gaochang Station and Cuntan Station in the testing stage; Figure S5: The decision tree visualizing plot in the RF_Step model at Cuntan Station. Table S1: The candidate predictive variables for runoff forecasting models; Table S2: Statistical metrics of 1-month-ahead runoff forecasting results of ML models for the Gaochang Station and Cuntan Station; Table S3: Statistical metrics of 1-month-ahead runoff forecasting results of univariate models for the Gaochang Station and Cuntan Station.

**Author Contributions:** Conceptualization, X.L. and L.Z.; methodology, X.L. and S.Z.; software, X.L. and L.L.; formal analysis, X.L. and Z.T. (Zhengyang Tang); resources, S.Z.; writing—original draft preparation, X.L.; writing—review and editing, L.Z., S.Z., Q.Z., Z.T. (Zhenyu Tang) and X.H.; All authors have read and agreed to the published version of the manuscript.

**Funding:** This work was supported by the Hubei Key Laboratory of Intelligent Yangtze and Hydroelectric Science Foundation (Grant Number: ZH20020001). We also acknowledge support by the National Key Research and Development Program of China (Grant Number: 2017YFA0603704), the Major projects of the National Natural Science Foundation of China (Grant Number: 41890824), the Excellent Young Scientists Fund, the Strategic Priority Research Program of the Chinese Academy of Sciences (Grant Number: XDA23040500), and the Youth Innovation Promotion Association, CAS (Grant Number: 2021385).

**Institutional Review Board Statement:** Not applicable.

**Informed Consent Statement:** Not applicable.

**Data Availability Statement:** The data that support the findings of this study were derived from the following resources available in the public domain: monthly streamflow, https://data.cma.cn/ (accessed on 20 June 2022); monthly global circulation indexes, http://cmdp.ncc-cma.net (accessed on 20 June 2022); air pressure, average temperature, maximum temperature, minimum temperature, relative humidity, wind speed and daylight hours, https://data.cma.cn/ (accessed on 20 June 2022).

**Acknowledgments:** We thank the anonymous reviewers for their constructive feedback.

**Conflicts of Interest:** The authors declare no conflict of interest.

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
