# Peer review of "Predicting Monthly Runoff of the Upper Yangtze River Based on Multiple Machine Learning Models"

_sustainability, doi:10.3390/su141811149_

Round 1

Reviewer 1 Report

1. Introduction section needs to be supplemented with thorough and relevant research background on the application of copula entropy in hydrological research and mid-to-long-term runoff prediction.

2. The choice of the numerous candidate predictive variables was not explained clearly. The authors need to add more previous research to further demonstrate the rationality of these candidate predictive variables, including the global circulation indexes, surface meteorological data, and the antecedent runoff.

3. In the Discussions section, the authors are suggested to highlight important findings and include afterthought of this work clearly, especially the superiority and suitability of the LSTM_Copula model in monthly runoff prediction.

4. On page 2, it is a little confusing with the acronyms, MAPE, RMSE, NSE, and R. Please complete the full name of the acronyms in the Abstract.

5. There is a slight difference in the format of Figure 8 and Figure 9. The latter has a dashed line where the vertical axis scale is 0. Please harmonize the formatting of the figures.

6. The manuscript needs careful editing and particular attention to English grammar, spelling, and sentence structure.

Reviewer 2 Report

1-      The title can be revised to better show the research content.

2-      Why did you use Pearson's Correlation Coefficient for variable selection? How about other methods? Pearson's Correlation Coefficient only considers linear relationship between variables.

3-      The authors should explain about what they did in modeling process e.g. how they calibrate each model separately.

4-      Why 12 lag times were used? It should be justified.

5-      It is highly suggested that the authors compare the models based on their computational cost. A comprehensive discussion is sufficient if the quantitative comparison is not possible at this stage.   

Reviewer 3 Report

Dear authors,

I have first run the similarity scores for this manuscript and found out 28% of the text are copied from other sources.

In fact, it has 4% from number 1 and 2. Just copied sections.

This is very unacceptable. 

Therefore, you are encouraged to work on your plagiarism scores and then to re-submit this. 

Round 2

Reviewer 3 Report

Predicting monthly runoff of the Upper Yangtze River based on multiple Machine Learning models.

I have rechecked the manuscript for similarities and found to be it is now around 22%. I understand that there are some elements in various repositories as explained by the authors and happy to continue with their paper. 

To me, the authors have substantially improved their manuscript with the comments of other 2 reviewers and therefore, my technical concerns are limited. 

One important thing; the authors should tell the reader, why they have considered monthly streamflow instead of daily streamflow? To me, monthly runoff is important for some of the applications (like hydropower development) but the daily streamflow  is the most required one for floods. Therefore, context of application is required. 

"In this study, we aimed to enhance the precision of monthly streamflow prediction and to analyze the role of multi-variable inputs in monthly runoff prediction"

Figure 5 needs more physical interpretations.

Figure 6c is unreadable.

What is your idea on Cascaded-ANFIS (fuzzy logics) for prediction?Cascaded Adaptive Network-Based Fuzzy Inference System for Hydropower Forecasting
